# An Integrative Review of Interprofessional Collaboration in Health Care: Building the Case for University Support and Resources and Faculty Engagement

**DOI:** 10.3390/healthcare8040418

**Published:** 2020-10-22

**Authors:** Deborah Witt Sherman, Monica Flowers, Alliete Rodriguez Alfano, Fernando Alfonso, Maria De Los Santos, Hallie Evans, Arturo Gonzalez, Jean Hannan, Nicolette Harris, Teresa Munecas, Ana Rodriguez, Sharon Simon, Sandra Walsh

**Affiliations:** 1Department of Graduate Nursing, Florida International University, Miami, FL 33199, USA; sawalsh@fiu.edu; 2Department of Undergraduate Nursing, Florida International University, Miami, FL 33199, USA; mflower@fiu.edu (M.F.); delossan@fiu.edu (M.D.L.S.); artgonza@fiu.edu (A.G.); jhann001@fiu.edu (J.H.); simonsh@fiu.edu (S.S.); 3Department of Communication Science and Disorders, Florida International University, Miami, FL 33199, USA; aalfano@fiu.edu; 4Department of Anesthesia, Florida International University, Miami, FL 33199, USA; falfonso@fiu.edu (F.A.); hevans@fiu.edu (H.E.); 5Department of Athletic Training, Florida International University, Miami, FL 33199, USA; nstallwo@fiu.edu; 6Department of Physical Therapy, Florida International University, Miami, FL 33199, USA; tmunecas@fiu.edu; 7Department of Occupational Therapy, Florida International University, Miami, FL 33199, USA; acaldero@fiu.edu

**Keywords:** interdisciplinary, interprofessional, transdisciplinary, multidisciplinary, collaborative, health care team, teamwork

## Abstract

*Background*: In 2010, the World Health Organization issued a clarion call for action on interprofessional education and collaboration. This call came forty years after the concept of interprofessional collaboration (IPC) was introduced. *Aim*: To conduct an integrative review of interprofessional collaboration in health care education in order to evaluate evidence and build the case for university support and resources and faculty engagement, and propose evidence-based implications and recommendations. *Search Strategy*: A literature search was conducted by an interprofessional faculty from a college of nursing and health sciences. Databases searched included CINAHL, Medline, Eric, Pubmed, Psych Info Lit., and Google Scholar. Keywords were interdisciplinary, interprofessional, multidisciplinary, transdisciplinary, health care team, teamwork, and collaboration. Inclusion criteria were articles that were in the English language, and published between 1995 and 2019. *Review Methods*: Thirteen interprofessional team members searched assigned databases. Based on key words and inclusion criteria, over 216,885 articles were identified. After removing duplicates, educational studies, available as full text were reviewed based on titles, and abstracts. Thirty-two articles were further evaluated utilizing the Sirriyeh, Lawton, Gardner, and Armitage (2012) review system. Faculty agreed that an inclusion score of 20 or more would determine an article’s inclusion for the final review. Eighteen articles met the inclusion score and the data was reduced and analyzed using the Donabedian Model to determine the structure, processes, and outcomes of IPC in health care education. *Results*: Structure included national and international institutions of higher education and focused primarily on undergraduate and graduate health care students’ experiences. The IPC processes included curricular, course, and clinical initiatives, and transactional and interpersonal processes. Outcomes were positive changes in faculty and health care students’ knowledge, attitudes, and skills regarding IPC, as well as challenges related to structure, processes, and outcomes which need to be addressed. *Implications/Recommendations/Conclusions*: The creation of a culture of interprofessional collaboration requires a simultaneous “top–down” and “bottom–up” approach with commitment by the university administration and faculty. A university Interprofessional Strategic Plan is important to guide the vision, mission, goals, and strategies to promote and reward IPC and encourage faculty champions. University support and resources are critical to advance curricular, course, and clinical initiatives. Grassroots efforts of faculty to collaborate with colleagues outside of their own disciplines are acknowledged, encouraged, and established as a normative expectation. Challenges to interprofessional collaboration are openly addressed and solutions proposed through the best thinking of the university administration and faculty. IPC in health care education is the clarion call globally to improve health care.

## 1. Introduction

In 2010, the World Health Organization (WHO) [1] issued a clarion call titled: Framework for Action on Interprofessional Education and Collaborative Practice. The WHO framework came forty years after the concepts of interprofessional education (IPE) and interprofessional collaboration (IPC) were introduced [2]. The concept of interprofessional education (IPE) is as an essential combination of knowledge, attitudes, values, skills, and behaviors that make up collaborative practice. IPE allows for team-based problem solving and promotes the best thinking of health professionals in offering quality health care [1]. IPC represents the interaction between the professionals from various disciplines who share the same goals with collective action [3]. The Canadian Interprofessional Health Collaborative [4] describe IPC as the process of developing and maintaining effective working relationships to obtain optimal health outcomes. 

Currently, health care systems are faced with significant issues of preventable mortality and morbidity, increasing medical errors, inadequacies in costly and fragmented systems of care, as well as lack of patient-centered care [5,6]. Thus, IPE and IPC are deemed critical to clinicians, researchers, professional groups, and government [7,8]. Institutions of higher learning are being called to lead transformational change in health care education beginning with a review of the curricula and educational outcomes of the various health disciplines and areas where IPC naturally align. The WHO [1] report emphasized that educational and health care systems must coordinate to develop educator and curricula mechanisms which support IPC.

To address implementation issues, the WHO [1] report also emphasized the importance of institutional support and working culture mechanisms, such as communication strategies, conflict resolution policies, and shared decision making. Additionally, issues were discussed related to the need for building space and facilities that accommodate IPC. The WHO [1] report suggested that activities conducted during formal education programs could provide future health care providers with critical knowledge. Such knowledge would hopefully motivate students to provide interprofessional services when students become part of the health care workforce. 

The WHO [1] framework not only identified mechanisms to shape successful teamwork but also outlined actions to be applied in academic and health care systems. In accordance with the Interprofessional Education Collaborative (IPEC) [9], faculty in Colleges of Health Professions were encouraged to develop core competencies for collaborative practice in all professional curricula. Such core competencies would embed essential content of interprofessional communication, patient-family centered care, role clarification, collaborative leadership, and conflict resolution. Students who engage in IPE are more likely to collaborate and implement interprofessional health care [10,11,12,13,14], with the proposed ultimate outcomes as improvement in the experience of health care, and quality of health care with a reduction in cost [1,8]. 

In addition, an Institute of Medicine Report (IOM) provided an interprofessional learning continuum model to implement and evaluate interprofessional educational efforts focused on preparing future health practitioners with interprofessional collaboration skills in the workplace. The WHO framework and the IOM model address the growing body of evidence that well-planned interprofessional learning is the antecedent of patient-centered, cost-effective, efficient, safer, timelier, and more equitable health care [1,8,15,16]. Through interprofessional education and training, health professionals may come to recognize shared values and codes of conduct, and move from a sense of independence to interdependence to provide patient-centered, holistic health care [16]. A paradigm shift in the education emphasizes the critical fit between interprofessional education and real-world interprofessional collaborative practice across health care settings [16,17]. 

The aim of this integrative review was to evaluate evidence related to interprofessional education in health care education and identify structures, processes, and outcomes that may serve to build the case for university support and resources and faculty engagement, and propose evidence-based implications and recommendations. 

## 2. Review Methodology

Cooper’s [18] five-stage integrative review method, as modified by Whittemore and Knalf [19], was used to guide the review, including (1) problem identification (defining the problem); (2) searching the literature (data collection); (3) quality appraisal (evaluation of the data); (4) data analysis and interpretation (data abstraction); and (5) presentation of the results (data synthesis). 

### 2.1. Problem Identification

The interprofessional committee of a college of nursing and health sciences identified the need to evaluate evidence related to the outcomes of interprofessional collaboration (IPC) in health care education. The committee acknowledged that the strategic plans of universities often address the importance of IPC. However, moving the idea of IPC in health care education from the thought phase to the planning and implementation phases requires the buy in of several stakeholder groups including individual learners, educators, health care professionals, researchers and very importantly administrators. Furthermore, the resources and funds allocated to promote interprofessional initiatives are dependent on providing the evidence which supports not only the efficacy of IPC in terms of learning outcomes and health care outcomes, but of the context through which those outcomes are achieved. 

### 2.2. Searching the Literature

To identify articles relevant to the literature review, thirteen members from six health professions of the college interprofessional (IP) committee formed teams and conducted an initial search of the following databases: CINAHL, Medline, Pubmed, Eric, Psych Info Lit, and Google Scholar. Search keywords were: interdisciplinary, interprofessional, transdisciplinary, multidisciplinary, collaborative, health care team, and teamwork. Inclusion criteria were English language, and published from 1995 to 2019. Although the concept of interprofessional education can be found in the literature as early as the 1970′s, we speculated, based on knowledge of the health sciences, that research studies would most likely be published from 1995 and beyond.

Based on key words and inclusion criteria, over 216,885 articles were identified. After removing duplicates, educational studies that were full text were identified and reviewed first by title and then based on their abstracts. Sixty-three articles were uploaded into a Google drive database to be accessible for review by the entire team. Six systematic reviews were reviewed, as reference points, but 32 single studies were included for an in-depth appraisal (refer to Figure 1: Search Strategy). 

### 2.3. Quality Appraisal/Evaluation

The articles were then assigned to teams to evaluate the strength of the 32 studies using the Quality Assessment Tool (QAT) developed by Sirriyeh, Lawton, Gardner, and Armitage [20]. This tool ranked the studies on 14 criteria for both quantitative and qualitative studies and two additional criteria added to the score for mixed-methods studies. The scoring of each of the 14 criteria was between zero and three points with the range of scores from 0 to 42. Each member of the team individually scored their team’s assigned articles. The assigned team then met to discuss their individual scorings. If there was a discrepancy between scores, the criteria for scoring was discussed and the team members established inter-rater reliability of each article reviewed so that a single score could be cited on the QAT. 

After inter-rater reliability was achieved on the 32 educational articles, the IP committee met, and each review team presented the scoring of their assigned articles. The IP committee determined that educational research articles with a score of equal to or greater than 20 points would be included in the integrative review as this was the median value of the range of scores. Eighteen articles met the criteria for inclusion in the review having a score of 20 or greater. 

### 2.4. Data Analysis and Interpretation

Studies to be included in the review were then entered into a table which identified the article’s authors, year of publication, country, journal, title, study aim, design, sample/setting, educational intervention, findings/outcomes, and implications (refer to Table 1). 

To further reduce and analyze the data, Tavares de Souza, Dias da Silva, and de Carvalho [21] suggested to categorize the data based on a pre-determined conceptual classification. Accordingly, the Donabedian Model [22,23], which identifies the structure, processes, and outcomes, was used to analyze the selected articles regarding IPC in health care education. According to Donabedian [23], structure speaks to the setting or context, including the organizational characteristics, as well as to the attributes of the provider in which the service is provided. Structure also includes the physical facility and human resources. Process denotes the transactions between groups which may be classified as technical processes or interpersonal processes and contains all health care initiatives. Outcomes are the effects on various populations, such as changes in knowledge, behavior, attitudes, and sense of satisfaction or improvement in the quality of care. 

## 3. Presentation of the Results in Accordance with the Donabedian Model

### 3.1. Structure

Based on the 18 articles which met the inclusion criteria, structure was identified with regard to the country of origin, settings, participants, disciplines represented, and study designs.

With regard to the *country of origin*, of the 18 studies, ten were in the United States, two in Canada, and one each in Germany, Sweden, and one across Europe. The *settings* for 18 studies were in universities or public colleges, while one also involved military medics on a military base, and one study was conducted by 16 health professional institutions at a Student-Run Clinic Conference. *Study participants* included (1) undergraduate students who were in nursing programs or health science programs; (2) graduate students in the health professions; (3) military medics; (4) faculty members; and (5) administrative and academic staff. The vast majority represented were graduate students. *Disciplines represented* included undergraduate and graduate nursing students, medical students, and students and practitioners of occupational therapy (OT), physical therapy (PT), physician’s assistants (PA), respiratory therapists (RT), public health, dental, pharmacy, social work, nutrition/dietetics, rehabilitation, psychology, mental health, medical administration, family therapy, speech pathology, exercise, osteopathic students, chiropractors, physiotherapists, medical technicians, orthoptics, paramedics, military medics, and laboratory assistants. *Study designs* included three qualitative, exploratory case studies, four descriptive, correlational surveys, one retrospective secondary analysis, eight intervention studies, one mixed-methods descriptive study, and one simulation case study (refer to Table 1).

### 3.2. Processes

A review of processes, as described by the Donabedian Model [23], identified three types of interprofessional initiatives, specifically curricular, course, and clinical initiatives, as well as descriptions of various transactions or interpersonal processes (Refer to Table 2).

### 3.3. Outcomes

The outcomes of this integrative review have been identified as positive changes in knowledge, attitudes, behaviors, and skills associated with interprofessional collaboration in health care education, as well as identifying the challenges associated with structure, processes, and achieving positive outcomes (Refer to Table 3).

## 4. Discussion

A synthesis of the evidence regarding the *structure of IPC* in health care education highlights that faculty, and supporting staff, in universities and colleges, both in the United States and abroad, are conducting high-quality research studies regarding IPC in health care education. Institutions of higher learning are involving undergraduate and graduate students, across the spectrum of health care professions, as well as professionals already in practice, in response to the WHO [1] call for transformational change in health care education. An integration of high quality descriptive, correlational, intervention, qualitative, and mixed-methods research studies is important to create the science base related to IPC in health care education [4]. In the competitive world of academics, faculty emphasis on leading-edge concepts, such as interprofessional collaboration, may move the needle towards a keen interest of university administrations in the support and funding of interprofessional initiatives. This may be particularly true, if interprofessional collaboration is linked with other benchmarking criteria of institutions, such as occurs in state university systems. 

This integrative review has identified the *processes of IPC* in health care education with successful outcomes, specifically positive changes in knowledge, attitudes, and behaviors/skills of participants. This evidence may be used to build the case for university administration support and resources for curricular, course, and clinical initiatives, as well as faculty engagement. First, curriculum across the university should raise the concept of interprofessional collaboration to a high level of awareness. To do this, administrators can fund faculty to develop an independent interprofessional course, develop interprofessional modules within a curriculum, or make a curricular shift to fund interprofessional teaching of a course. Secondly, within courses in a college or department, administrators can call for the weaving of interprofessional objectives through each course, with associated course readings and assignments, or even the development of specific course modules in which content is tested. In any course, a case study methodology can be used to discuss the roles and competencies of the various health professionals and their expected contributions to the quality of care offered. Role playing the role of another professional provides a learning opportunity to change the lens from which a plan of care is developed. Thirdly, administrators can establish relationships with external clinical agencies to provide student with learning opportunities to experience interprofessional collaboration, such as shadowing of differing health professions. Additionally, administrative funding of large-scale clinical disaster simulations, or cased-based educational simulations in a clinical simulation lab are opportunities which may be planned to engage different health care students (i.e., medicine, nursing and health sciences, public health, psychology). This may pave the way for a cultural shift and normative expectations of the university regarding interprofessional collaboration. The involvement of an interprofessional faculty across departments or colleges showcases the transactions and interpersonal, as well as interprofessional processes as reported by the studies of the integrative review (Refer to Table 1 and Table 2). Faculty role modeling of coordination, communication, cohesion, team planning, decision making, and performance feedback is an important learning strategy for undergraduate and graduate students and junior faculty.

In 2008, Berwick, Nolan and Whittington [42] proposed a Triple Aims framework to evaluate outcomes of interprofessional health care teams, specifically the evaluating the experience, access to, and the cost of health care. Using this framework, Brandt, Lutfiyya, King, and Chioreso [2] conducted a systematic review of IPC and IPE with 20 disciplines. Although the outcomes of quality and cost of health care could not be explicitly mapped to the Triple aims, the results indicated an impact on changes in readiness, attitudes and perceptions of health care providers regarding interprofessional collaboration. Brandt et al. [2] emphasized the importance of exploring further the processes of collaboration and measuring outcomes. 

Gathering university-wide data, obtained through varying testing methods, including quantitative tests, as well as qualitative data through observations, and participant narratives, can be used to measure the *IPC outcomes* of curricular, course and clinical initiatives, and the related transitions and interpersonal processes. This integrative review highlights the resultant positive changes in knowledge, attitudes, skills/behaviors of health care students, faculty, staff and other practicing health professionals regarding interprofessional collaboration, as was also reported by Brandt and colleagues [2]. Given the importance of evidence-based education and practice, the results of educational research studies, as presented in Table 1 and synthesized in Table 3 as positive outcomes, helps to build a case for university administrative support and resources to further interprofessional collaboration in health care education. However, the outcomes also highlight the challenges of interprofessional collaboration in health care education, related to structure, processes and outcomes, which need to be analyzed by the university administration in collaboration with university faculty and with acknowledgment of staff and student perceptions. Through their active engagement, there is an opportunity to create a cultural shift at a university and move toward the institutionalization and normalization of IPC in health care education. 

## 5. Implications and Recommendations

The value of diverse integrative reviews is that each may provide a varying lens with which to view the phenomenon of interprofessional collaboration in health care education and substantiate similar or differing implications. Through the lens of the Donabedian Model [23], the articles included in this integrated review were examined for structure, processes, and outcomes. Our goal was to provide evidence to build the case for university support and resources related to interprofessional collaboration in health care education, further generating valuable ideas and important implications for university administrations and engaging faculty across colleges and units. 

First, to promote successful interprofessional collaboration, what is needed is a simultaneous top–down and bottom–up approach that involves commitment from university and college administration, chairs and directors of departments and programs, faculty, staff, students, and colleagues from affiliated health care institutions [43]. Interprofessional initiatives must illustrate commitment to the advancement of each health discipline, allowing their ability to practice to their full extent of their scope of practice, yet with optimal team cooperation, and collaboration [43]. Transformative change in education and clinical practice involves interprofessional networking of colleagues within, across, and beyond the university and those within health care systems and agencies. 

Second, the Faculty Senate of a University may create a Senate Interprofessional Committee focused on interprofessional collaboration. This committee may be charged with developing an Interprofessional Strategic Plan, which includes clear vision and mission statements related to interprofessional collaboration in education, research/scholarship, and clinical practice/service, with identified and measurable goals, strategies, tactics, timeline, and products. With administrative support at the university and college levels, signature IP events may be hosted each year, with an invitation to renowned speakers, showcasing successful and exemplary IP initiatives that are occurring at the university and colleges, and conducting focus group discussions of event participants to answer strategic questions with regard to facilitating the processes of collaboration, while overcoming barriers or challenges. At the university or college level, funding may be offered to support formal collaborations for interprofessional research, educational, or practice/service initiatives. Centers for academic teaching may structure workgroups to learn strategies that promote interprofessional collaboration. In university online platforms, such as Blackboard or Canvas, an online non-credit course may be developed with support of administration to serve as a repository for documents, videos, PowerPoint presentations, lectures, discussion boards, and Linked in Learning or YouTube videos related to interprofessional collaboration. Eventually, this course may be developed as a professional development course for university/college students, faculty or staff and eventually be opened to the public for continuing education.

Third, at the college or unit levels, it is important to strategically map IP collaboration across the curriculum by reviewing existing courses, which may already emphasize interprofessional collaboration and practice, or be identified as a course in which common or shared competencies of health professionals provide a valuable venue for interprofessional teaching-learning strategies. All existing courses may be reviewed and modified with the intent to embed essential content of interprofessional communication, patient-family-centered care, role clarification, collaborative leadership, and conflict resolution, as well as including course readings, assignments, and assessment strategies that focus on interprofessional collaboration. 

Fourth, it may also be of value to develop an interprofessional course with joint teaching assignments by an interprofessional faculty. The course may extend beyond a didactic course to have a clinical component to promote the practice of interprofessional knowledge and skills in a real-world setting. The course potentially may include clinical assignments in which students may receive feedback from patients and families regarding the care offered by an interprofessional student team. In the proposed interprofessional course, there would be an opportunity to discuss important topics from an interprofessional, as well as discipline specific lens. For example, in the interprofessional course, one goal may be to develop a case study using a comprehensive, interprofessional assessment template. This template may offer the opportunity to recognize shared or unique competencies of various health providers. During initial classes, students may role play a different discipline than their own to appreciate their expertise. Students may participate in panel discussions of a case from an IP perspective, and reflect on personal experiences when a team approach is used to achieve a goal.

As suggested in multiple articles reviewed (Refer to Table 1) [24,25,26,27,28,29,30,31,32,33,34,35,36,37,38,39,40,41], environment is an important aspect in promoting IPC. Besides formal strategies to promote IPC in health care education, colleges or units may provide collaborative environments or learning space, such as shared lounges, shared classrooms, shared simulation labs, and the availability of classroom or community/practice setting for team activities. Department chairs and program directors may suggest opportunities for collaboration with colleagues within or external to the unit. Interprofessional collaborations on research submissions, academic publications or presentations, building online courses and websites, or constructing evaluation tools and exams can be opportunities to support collaboration. 

At the university, college and unit levels, it is important to identify individuals who champion IPC and who recognize the value of IPC and IPE in terms of their own professional development, and academic and health care outcomes. Equally important is to share best practices regarding IPC. This may take the form of creating a university online repository of research and best-practice articles published by leading organizations in IPE and IPC. Lastly, the university, colleges, and units may develop normative expectations related to IPC with documentation of IPC in the tenure and promotion process and the annual merit review process, as well as developing formal mechanisms to reward collaborative efforts.

## 6. Limitations to the Approach

The undertaking of an integrative review by a large team of faculty members is challenging as the experience and expertise of team members may vary. For example, members of the interprofessional committee may be faculty on a clinical track, a tenured-earning track or be tenured faculty with different or similar professional expectations set by the university or college administration. Team members may also have varying professional experience related to interprofessional collaboration, as well as experience with the review methodology, and expertise with library searches and search strategies. It is acknowledged that such limitations may result in the omission of valuable articles relevant to the review identified by colleagues well-versed in the literature regarding interprofessional collaboration. There are faculty team champions who have the knowledge to set up google drives and upload documents, while other team members have less technological savvy. In addition, an integrative review is a relatively long process of scholarship, which requires ongoing commitment and dedication to the review process. Limitations to the approach may also relate to the ability of team members to be flexible in the scheduling of multiple full team meetings, as well as individual group meetings, as the faculty team moves forward from conceptualization of the project, to accepting various administrative roles, participating in all aspects of the search methodology, documentation of results and analysis. Individuals involved in writing of the review article depend on timely feedback from all team members and a thorough critique and editing of the article to the point of journal submission. The limitations to the approach can be overcome by the ongoing commitment of each team member to the project and to each other as colleagues involved in a process of collaboration.

## 7. Conclusions

This literature review regarding interprofessional collaboration in health care has applied the Donabedian Model [23] lens of structure, processes, and outcomes to analyze the evidence and build the case for university support and resources and faculty engagement. Equally important, it has stimulated valuable ideas and important implications and recommendations for university administration and faculty across colleges and units to promote interprofessional collaboration in health care education. The essential message is that the creation of a culture of interprofessional collaboration requires a simultaneous “top–down” and “bottom–up” approach. When valued by the university administration, opportunities can be created to promote and very importantly, reward collaboration in education, clinical practice, and research/scholarship. The university administration, in collaboration with the Faculty Senate, can lead the way by creating and supporting a Faculty Senate university committee, seeking the support of senate champions who are influencers of faculty across colleges, units, and departments. The grassroots efforts of faculty to collaborate with colleagues outside of their own disciplines are not remanded, and confined in silent silos, but rather are acknowledged, encouraged, and perhaps, established as a normative expectation. Challenges to interprofessional collaboration are openly addressed and solutions proposed through the best thinking of the university administration and faculty, as well as staff and students. Interprofessional strategic planning, involving all stakeholders, has the potential to serve as a catalyst for the synergistic interface of interprofessional collaboration in health care education, clinical practice and research/scholarship. Interprofessional collaboration may become a critical force in the transformation of global health care. 

## Figures and Tables

**Figure 1 healthcare-08-00418-f001:**
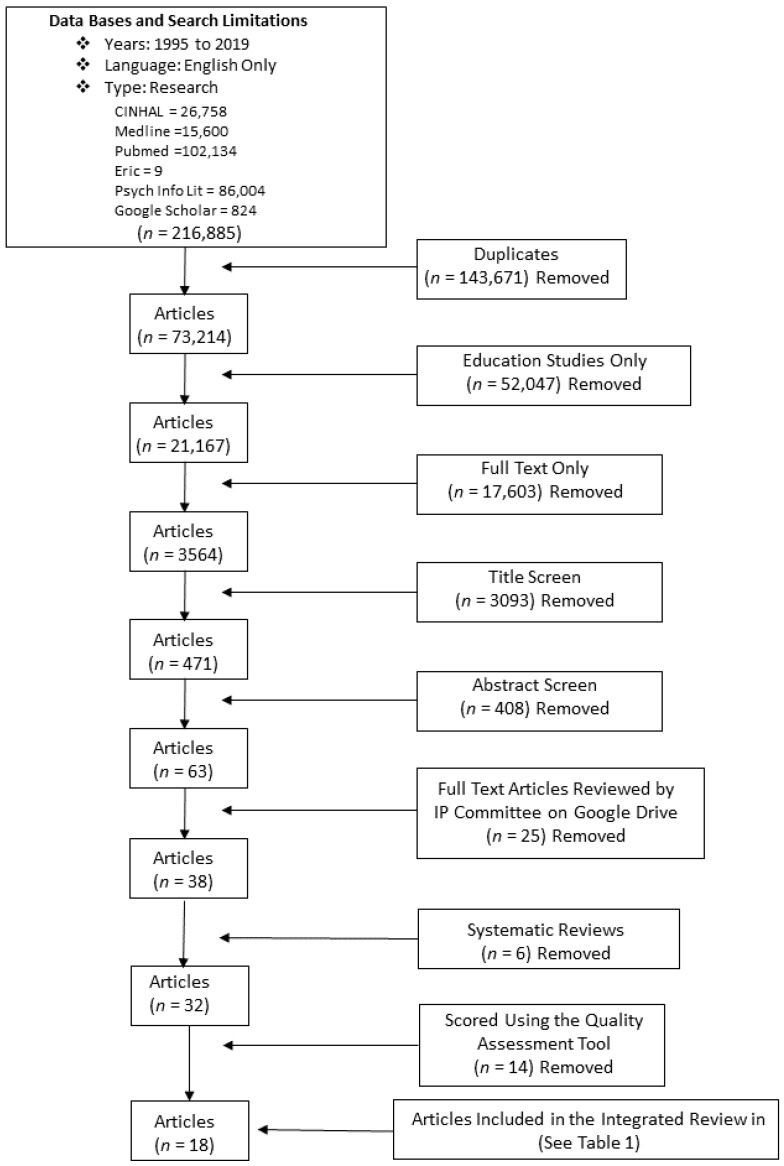
Search Strategy.

**Table 1 healthcare-08-00418-t001:** Articles Selected for Inclusion in the Integrative Review

	Authors, Year, and Country	Study Aim	Study Design	Sample and Setting	Intervention	Findings/Outcomes	Implications
Article 1	Murray, B., Judge, D., Morris, T. and Opsahl, A. (2019). US [24]	To describe how a disaster response simulation can be utilized as an experiential learning technique fostering interprofessional collaboration.	Evaluation research	38 sophomore nursing students from the traditional baccalaureate nursing (BSN) program, 23 junior nursing students from the BSN program, 16 students from an associate degree program, 4 paramedic students from the community college and 14 military medics in training. Simulation took place on a military base in the mid-west US.	Disaster simulation based on the International Nursing Association for Clinical Simulation and Learning Standards for Best Practice.	Student survey of learning objectives indicated that the highest ranked objective was collaboration with other disciplines and health care providers. The lowest scored objective was developing a holistic plan of care addressing the individual needs of the simulated patient.	Simulation engaged students in critical thinking while allowing practice in safe environments. Planning and execution of the event between three institutions addressed the goal to improve interprofessional education.
Article 2	Chen, K, Kruger, J., McCarther, N., and Meah, Y. (2019). US [25]	To create a space for communication between participants in Student-Run Clinics to discuss shared challenges and possible solutions, and motivate collaborative practice on practice changing ideas.	Post-test survey	23 participants representing 16 institutions and 5 professions, including medical, pharmacy, physical theory, nurse practitioner, and undergraduate pre-medical students.	Novel, abridged hackathon workshop at a conference piloted by the Society of Student-Run Free Clinics.	Proposals developed addressed wait times, follow-up, quality improvement, patient education, community engagement and interprofessional collaboration. Twenty-one of participants would likely implement an idea discussed during the event; 17 participants responded favorably to collaboration.	The abridge hackathon encouraged inter-clinic and interprofessional engagement around solving shared programs. Though participants their likeliness to collaborate with other clinics after the event, longer-term benefits of this educational event is needed.
Article 3	Mahler, C., Schwarzbeck, V., Mink, J., and Goetz, K. (2018) Germany [26]	To report on and gain insight in the students’ perspective on interprofessional learning in general within a new science program.	Qualitative, exploratory case study	49 bachelor of science program “Interprofessional Health Care” at Heidelberg University participated. Data was collected through focus groups, semi structured guideline, and audio and video recordings with transcriptions.	Interprofessional education and learning during the first two semesters.	Interprofessional learning is perceived positively by the students at this early stage in their studies and was associated with benefits and challenges. A positive interprofessional atmosphere within the group was perceived and the wish to engage more with medical students was stated.	Recognizing prejudices and stereotypes prevalent in the health system, students will discover ways to overcome these. They will encounter their coworkers in an open manner and will be able to develop a better mutual understanding.
Article 4	Achkar, M., Hanauer, M., Colavecchia, C., and Seehusen, D. (2018) U.S. [27]	To identify the prevalence and format of, the participants in, and the barriers to IPE; to examine the goals and assessments of IPE experiences; and to explore potential IPE models for programs that do not currently use IPE.	Online survey questionnaire via RedCap	233 graduate medical education program directors.	Evaluation of various interprofessional education experiences.	The median number of hours of IPE was 60 hrs. Barriers were: 1) time for teachers (54.4%), 2) time for residents (51.5%), 3) financial support (33.6%), 4) space to host activities (30.7%), and 5) faculty buy-in (25.2%). Reasons for benefits to IPE were: “to improve collaboration” (92.2%), 2) “to improve communication” (87%), 3) “to improve patient safety” (82.6%), 4) “to improve health care quality” (79.1%), and 5) “to improve attitudes towards teamwork” (71.3%). Outcomes were: “skills for working on an interdisciplinary team” (53.9%), “satisfaction with the learning experience” (49.6%), “attitude towards interdisciplinary teamwork” (44.4%), “content specific knowledge” (32.2%), and “attitudes towards specific content” (33.9%).	Future research should examine how programs have addressed the barriers to IPE. A qualitative study, interviewing program directors, could study programs as they implement IPE to understand how such barriers are overcome. The findings of future research could be shared with GME programs interested in implementing IPE to begin a dialogue.
Article 5	Peterson, J., Brommelsiekv, M., and Amelung, S. K. (2017) U. S. [28]	The aim is to prepare health care providers that are capable of functioning in interprofessional clinical practice (IPCP) teams to provide compassionate, high quality care for veterans and military families.	Mixed-methods (quantitative/qualitative) educational interventional study	US-VHA health care facility. Health professional students, advanced practice nursing, pharmacy, clinical psychology and social work students at a US Midwestern university.	An 8 week IPE immersion course that included military culture, behavioral and physical health disorders common among veterans, and all related treatments. Faculty-led discussions with students in IPE teams used veteran-focused case studies and standardized patients. Data sources included quantitative surveys, Knowledge Assessment Tool, qualitative reflection, and focus groups.	At baseline, students showed high readiness for interprofessional learning. From pre- to post-course, a significant increase in knowledge of course curriculum and an increase in their perceived value of a team approach to providing care. Post-course, students reported high levels of communication, cooperation and collaboration among team members. Faculty articulated the benefits and modeled interprofessional collaboration with other course faculty. Themes from focus groups and reflection questions included Roles and Responsibilities, Teams/Teamwork, Cultural Understanding, Patient Advocacy, and IPE and Professional Education with an increased understanding and skillset for each.	IPE and team building helped health professional students to value each other’s contributions, communication, and collaboration to improve care provided to veterans. Students had varying academic levels allowing them to learn from one another. Working on patient case studies in interprofessional groups allowed them to improve their assertiveness and confidence in interacting with other professionals.
Article 6	Reed, C., Garcia, L.I., Slusser, M.M., Konowitz, S., and Yep, J. (2017) U.S. [29]	Link university level Essential Learning Outcomes (ELOs) related to Ethical Reasoning with Program Learning Outcomes (PLOs) and Student Learning Outcomes (SLOs), and the Interprofessional Collaborative Practice Core Competency of values and ethics in an introductory baccalaureate-level health science (BSHS) course.	Rubric was developed to evaluate the application of learning expectations and objectives and to measure their attainment against a consistent set of criteria. Narrative analysis of BSHS papers	94 Baccalaureate-level health sciences students.	Students were given a case study that required a multidisciplinary health care team to make an ethical decision. Students assumed the role of a specific member of the health care team. Using the case study, they explored the relationship of values and ethics to making a health care decision which involved identifying personal value, resolve conflicts, and develop one resolution that satisfied all members. Each group presented their findings and each student wrote an individual scholarly paper.	Results indicated that the majority of students achieved desired course and program outcomes related to ethical decision making. The course level objective related to the core competency of values and ethics was fully met. Most students achieved the skilled level of the university ELO mid-way through program at the end of the second Introduction to Health Science course. Role playing and reflection achieved desired learning outcomes. Using a rubric to score student’s competency when analyzing papers on ethical decision making provided an accurate evaluation of student learning that can be used in determining how well university level, program-level, and course-level learning outcomes were achieved.	Given students’ level in program, program goals related to values and ethics were adequately achieved. These students have one more core program course in which values and ethics are included as an objective. Evidence from this assessment project suggests that students should fully achieve program outcomes in values and ethics by completion of the final core course. Our results
Article 7	Shagrir, L. (2017) Europe [30]	Examine how higher education-based teacher educators perceive the issue of collaboration with their colleagues; investigate the nature and character of their collaborations; and examine what they acquire as a result of these collaborations.	Survey	The questionnaire was sent to 31 faculty members at an institution; 23 questionnaires were received, with 21 from women. Eighteen of the respondents have a Ph.D. and five have M.A. degrees. All arelecturers and some hold management roles simultaneously such as head of programs or units. Respondents’ experience and service as educators covered a wide range: nine of them had up to 10 years of experience, three had 11–15 years, six had 15–20 years, and four had over 20 years.	No intervention.	Collaboration with colleagues is perceived as an important component of their professional life and academic development. Preference for refraining from collaboration with inexperienced colleagues or those at the start of their professional journey. A total of 95.7% strongly agreed that it is possible to promote new initiatives and ideas through collaboration; 91.3% believe that professional and academic development can be advanced through collaboration; (78.3%) agreed that it is important for academic leaders to encourage and promote collaboration among faculty members, and that collaboration should be a part of the criteria in evaluation processes (65.2%).	Academic leaders should encourage interprofessional collaboration given that professional development and academic development is enhanced among faculty.
Article 8	Nagge, J., Lee-Poy, M., and Richard, C. (2017) Canada [31]	The goal of the event was to build interprofessional competency in the areas ofcommunication, collaboration and role clarification for medical and pharmacy students. This study was designed to evaluate self-reported changes in these domains using a validated pre–post-survey instrument.	Post-intervention survey	118 pharmacy students and 28 medical students at a Canadian university.	Half-day in-person event consisting of: a patient-interview station, a reflective interprofessional communication discussion, and a prescribing station.	Intervention appeared to have the strongest effects in category of collaboration (roles and responsibilities/collaboration/collaborative patient-/family-centered approach), while the least robust effects were noted in the conflict management/resolution category. The event led to significant improvement in all 20 items measured by the instrument. Results suggest that this activity was an effective IPE experience that met the objectives.	Planning and executing meaningful IPE activities requires investment of significant time and resources. Strong and consistent improvement of scores suggest a framework for pharm and med school training to move from siloed ed experiences to synergistic learning opportunities and lends support to the decision to make it an annual event.
Article 9	Hoffman, S., and Harnish, D. (2017) Canada [32]	To design, execute and evaluate the effectiveness of a mandatory IPE initiative targeting students in their first year of a general undergraduate health science education program.	Pre-test post-test design	162 Bachelor of health science (BHSc) students (99.4% response rate).	Three components tested in groups of eight students: (1) an introduction; (2) a stereotypes exercise; and (3) discussion of one of three patient case studies.	Based on a two-part questionnaire, which was developed based on the Modified Kirkpatrick’s Model of Educational Outcomes for IPE (Freeth et al. 2002), the results demonstrate a profound positive change in attitudes, interests, and knowledge among participating students.	Based on the results, mandatory IPE for pre-health professional students is certainly merited, yet additional research needed.
Article 10	Renschler, L., Rhodes, D., and Cox, C. (2016) U.S. [33]	To evaluate whether short-term interprofessional events or long-term interprofessional programs have greater impact on students’ attitudes towards interprofessional teamwork.	Pre–post-survey design	148 students enrolled in an osteopathic medical school and health sciences students participated in a one-semester (short) interprofessional program and 159 students participated in a two-semester (long) interprofessional program.	Interprofessional short program consisted of 2 h, large-group, evening orientation session followed by either one semester of geriatric home visits; or two semesters of geriatric home visits.	Based on the Attitudes Toward Health Care Teams Scale (ATHTC) and the Team Skills Scale (TSS), the results indicated that health sciences students rather than the medical school students showed significantly improved their attitudes towards interprofessional collaboration.	Changes in attitudes toward interprofessional education and collaboration is different among professions. The culture in medical schools may negatively influence the perception of the value of collaboration.
Article 11	Townsend, T., Pisapia, J., and Razzaq, J. (2015) U.S. and UK [34]	The aim of this study is to describe actions designed to foster interdisciplinary research efforts at a major university in the UK. (1) What are the perceptions of administrators and academic staff of the nature and benefits of interdisciplinary research? (2) How is interdisciplinary research, at the university, college, and school level, organized, led and supported?	Descriptive mixed-methods case study approach	127 academic staff responded to the survey and 25 interviews with heads of colleges, schools, research coordinators and teams.	No intervention.	Most respondents (84%) were actively involved in interdisciplinary research (IDR); 71% recognized the need for good leadership, but only 27% felt that this was being offered at the college level and 45% felt that university systems were too cumbersome. Responses showed that 47% of IDR team members and 53% of team leaders were undecided about colleges’ success in introducing IDR. Support for IDR involves training from the college to the project level.	IDR involves both a top–down approach by administration and a bottom–up approach by faculty. Interdisciplinary research and teaching require new policies and structures.
Article 12	Arenson, C., Umland, E., Collins, L., Kern, S., Hewston, L., Jerpbak, C., Antony, R., Rose, M., and Lyons, K. (2015). U.S. [35]	To describe the implementation of a required longitudinal IPE program relying on lay persons as educators; to identify short-term process outcomes for continuous curriculum improvement; and to conduct mid-range longitudinal evaluation of impact on student attitudes toward chronic illness care and IPE, understanding of the roles of professional team members and patient-centered care.	Mixed-methods descriptive study	577 students who were all entering medical, traditional baccalaureate nursing, OT and PT programs. Pharmacy and couple and family therapy (CFT) students were added in the second year.	A 2 year required interprofessional curriculum.	Results of t-tests showed significant improvements in IPE attitudes from baseline to the end of year two in each program. A major benefit is the collaboration that develops within student teams. Students’ written reflections revealed emergent awareness of and respect for the scope, rigor, and demands of their fellow team members’ courses of study and practice.	Students receive a rich education in what really matters to patients engaged in health care. Ongoing longitudinal evaluation will document how these early lessons are sustained to inform future practice.
Article 13	Borrego, M., Boden, D., and Newsander, L. (2014) U.S. [36]	To explore the effect of targeted federal funding on change in interdisciplinary graduate education.	Exploratory, conducted in two sequential phases: analysis of funded proposals and a descriptive case study of two institutions	114 funded National Science Foundation (NSF) Integrative Graduate Education and Research Traineeship (IGERT) proposals were reviewed.	No intervention.	Only 26 of the 114 IGERT grants mentioned plans for institutionalizing or continuing interprofessional efforts when funding was completed. The focus was on developing new interprofessional courses or certificates in graduate education. A total of 57 proposals mentioned creating a culture of interdisciplinary research and training, interdisciplinary thinking, the creation of interdisciplinary graduate courses, and increased communication across interdisciplinary groups.	Provides examples of changes in policies and cultural expectations. The study can inform future program evaluations and funding policy focused on institutional change. The research highlights the role of both structure and cultural norms evident in organizational change.
Article 14	Ekmekci, O. (2013) U.S. [37]	To explore how the integration of interprofessional components into health care curriculum impacts professional stereotyping and collaborative behavior in care.	Simulation case study	The sample included 1000 health care students (medical; registered nurse; physician assistant; physical therapy; radiation therapy program) of a university.	Half of the students (i.e., 500) completed a curriculum with no IPE component. The other 500 students completed the same curriculum in which 25% of the courses being offered had IPE components embedded. Upon completion, mean scores representing tendency for stereotyping were captured for all 1000 students completing the IPE and Non-IPE curricula.	The tendency for stereotyping was significantly lower (*p* < 0.001) for students attending curriculum containing IPE components, as compared to students attending curriculum without an IPE component. The team mean scores for collaborative behavior was significantly higher for those exposed to curriculum with interprofessional content.	The authors suggest that less stereotyping and greater collaboration in health care delivery teams, in turn, could result in improved outcomes, such as greater patient satisfaction, higher quality of care, more effective clinical treatment and more extensive information sharing.
Article 15	Hylin, U., Lonka, K., and Ponzar, S. (2011) Sweden [38]	Investigate health care students evaluation of interprofessional clinical training in relation to their professional study.	Pre- and post-test	369 students (40 OT, 85 medical, 52 physiotherapy, and 192 nursing) in a Swedish University Hospital.	Mandatory 2 week interprofessional education course with teams of students working together and providing patient care. Educational goals include developing own professional role, gaining knowledge about other professions, and increasing skills in communication and teamwork.	Based on the Conceptions of Learning and Knowledge Questionnaire, students preferred a collaborative-constructivist approach to learning. All students improved in their own professional role mean scores, as well as knowledge about other professional roles.	All students, regardless to their approach to learning, highly valued the interprofessional training in clinical practice.
Article 16	Miers, M., Rickaby, C., and Clarke, B. (2009) UK [39]	To promote collaborative learning with a view to developing skills for collaborative working.	Each university-based module adopted an inquiry-based learning approach and lasted six weeks, with assessments submitted up to eight weeks after the end of the module engagement period	Students from nursing, social work, OT, PT, mental health, radiotherapy and diagnostic imaging were invited to participate in the research program. Data on student experience was collected through observations, interviews and focus groups. Data on student learning was collected through interviews and analysis of completed assignments.	Students from different professions worked together in small groups to complete several inquiry-based learning cycles and organizing a group presentation at the end of each cycle.	This study showed that students learning in interprofessional groups were able to gain knowledge of group dynamics and awareness of their own personal skills. Analysis of assignments showed development of cognitive skills, moving from description to synthesis.	Faculty has committed to integrating interprofessional collaborative skills into professional curricula and has adapted methods of delivery and assessment in order to address resource constraints and student concerns about workload and professional mix.
Article 17	Priest, A., Roberts, P., Dent, H., Blincoe, C., Lawton, D., and Armstrong, C. (2008) UK [40]	To explore interprofessional attitudes arising from shared learning in mental health.	One-year pilot test followed by 2 year full project with different cohorts of students	38 students who are clinical psychology trainees and mental health nursing students, BSN students, and mental health volunteers at a university.	Structured interprofessional learning program in mental health, including experiential and creative group work activities, problem-based learning with clinical vignettes and ask the expert panels.	Based on the Readiness for Interprofessional Learning Scale, together with other quantitative and qualitative elements, the results show an increase in clarity regarding roles, approaches and resources and how to collaborate in practice.	In mental health, shared learning among team members is important within both educational systems and in clinical practice.
Article 18	Florence, J., Goodrow, B., Wachs, J., Grover, S., and Olive, K. (2007) U.S. [41]	To assess career choices, practice locations, and attitudes of Community Partnerships for Health Professions Education Program (CPP) graduates compared to traditional graduates at an East Tennessee State University (ETSU).	Survey	84 CPP graduates including medical students, bachelor of science in nursing students and public health students were compared to a matching cohort of traditional ETSU students on practice locations and careers, and incorporation of interdisciplinary philosophies in practice and attitudes toward professional preparation.	Three-year longitudinal curriculum including theoretical, conceptual, and practice elements of medical, nursing and public health students incorporated into an experiential, inquiry-based and service-learning program.	CPP graduates significantly had greater interest in program outcomes, and rated their preparedness to work on interdisciplinary teams as significantly higher.	Students trained in rural, community interdisciplinary settings are more likely to select to serve in rural areas upon graduation. It is suggested that an interdisciplinary communications course for health care incorporated into the traditional curriculum.

**Table 2 healthcare-08-00418-t002:** Processes Related to IPC in Health Care Education.

**Type of Interprofessional Initiatives**	Curricular Initiatives Institutional based curriculum including teamwork in a rural setting community partnership program. Incorporation of philosophies and attitudes toward professional preparation across the curriculum. Curriculum building on a senior mentor program over two years. IPE strategies within the curriculum including classroom learning, team-based learning, web-based learning, simulation, and 60 clinical hours with an IP team. Modular interprofessional curriculums. Eight-week IPE immersion program including hybrid approach of didactic and online learning, with interprofessional classroom, clinical case study, and use of standardized patients with reflection questions. Course Initiatives Secondary analyses and retrospective chart review regarding attributes and opportunities for IPE leaning. Use of simulation within an integrative course. Two-week mandatory interprofessional IPE course. Educational intervention over one or two semesters in a senior citizen independent living community including teamwork, patient safety, physical-psycho-social team-based assessment and long term and short-term exposure to IPE. Interprofessional learning modules with integrative exercises, self-directed research, problem -based, and collaborative, community education home visits. Case study approach with five de-identified patients. Two sessions in year one and two using self-directed work groups, case vignettes, and reflection on how IPE impacts clinical practice. Interprofessional modules to learn collaboration, communication and teamwork skills. Qualitative case study to identify students’ perceptions of interprofessional learning based on focused group interviews. Use of an ethical case study and role-play based on identification of own personal and professional values with resolution of ethical conflicts. Clinical Initiatives Half-day interprofessional education event with stations on communication, patient interviews and prescribing. Pre-test/post-test after single verses immersion IP experiences. Self-completed questionnaire to access faculty perceptions of IP collaboration. Pre- and post-test disaster simulation. Post-test survey of participants at Student-Run Clinic Hackathon, which is an event that promotes team-based innovation in a short time frame with the use of design thinking principles (MIT Hacking Medicine, 2019).
**Transactions and Interpersonal Processes**	Coordination, communication, cohesion, problem solving, planning, decision making, critical thinking, application of theoretical knowledge, social relations, performance feedback, and conflict management.

**Table 3 healthcare-08-00418-t003:** Positive Outcomes and Challenges Related to IPC in Health Care Education.

**Positive Outcomes Related to Knowledge, Attitudes, and Behavior/Skills Regarding IPC in Health Care Education**	**Changes Related to Knowledge** Sense of greater preparation and knowledge regarding working on interdisciplinary teams Increase in the knowledge and positive attributes of IPE Recognized the value of conceptual frameworks which address academic disparity Acquisition of interprofessional knowledge Increased self-awareness and understanding of health professional’s roles with increase in confidence to transfer collaborative learning experience to practice settings Increased knowledge of other professions including scope of practice, terminology and professional overlap Increased learning about theories related to group work Greater understanding of ethical issues and value in improving IP competencies **Changes Related to Attitudes** Less stereotyping Greater interest in working in collaborative community settings Increased student perception of a collaborative, constructive approach to IP learning with faculty and students Increased attitudes toward team values Majority of students, regardless of approach to learning, highly valued interprofessional education Improved attitudes to teamwork and collaboration with increased clarity of interprofessional roles Significant improvement in attitudes across all disciplines Reflective logs gave greater insight in group processes and own collaborative skills with deeper analytical skills over prejudices Belief that teamwork is beneficial to patient care with higher sense of interprofessional collaboration Less fear of working with other professions Greater readiness for interprofessional learning and open mindedness **Changes Related to Behavior/Skills** Increased collaborative behavior Increased in student interactivity, and frequency of interaction with more explicit positive statements Increase in interprofessional clinical placements and health promotion learning Improved communication, teamwork, relationship building, and mutual respect Move from tribalism where groups wanted to stay with same discipline members to move to working with other disciplines over time Improved outcomes in collaboration, communication, patient safety, team skills, as well as knowledge regarding interprofessional content and satisfaction with the learning experience Communication, respect, learning to work as a team, learning roles of other professions were viewed as most important outcomes that enhanced the university experience Significant improvement in communication, collaboration, role clarification, patient-family-centered care, team function, and least effect on conflict management and resolution Significant improvement in team efficiency Improved cultural understanding and patient advocacy Learned to approach patient care from a different perspective and adjust their interactions 10 years to see institutionalization and normalization of IPE into graduate education following a grant funded project ➢ Regulative pillar—regularizing behavior with IP courses and certificate program ➢ Normative pillar—successful programs have vision statement and goals ➢ Cultural cognitive pillar—ways of making meaning of a new IP culture of education and research
**Challenges Related to Structure, Processes and Outcome Regarding IPC in Health Care Education**	**Challenges Related to Structure** Restrictions on time and changes in work load Some disciplines have restrictive boundaries Concern that academic success is defined by success in a discipline Importance of having a strong leader Need clearly defined curricular goals Challenge to having student health professions at different levels of knowledge **Challenges Related to Processes** Lack of clarity regarding terms: multidisciplinary, interdisciplinary, interprofessional, etc. Feeling of threat of another discipline taking control Challenge to develop a common language across professions Need to spend more time for participants to get to know each other and create a group atmosphere with exchange of ideas and shared concerns Power struggles between interprofessional event planners Conflicts with scheduling Request for a higher level of participation from medical students Requested more opportunities for shared interaction **Challenges Related to Outcomes** Medical students and osteopathic students scored lowest on collaborative constructive scales; low collaborative scores associated with less satisfaction of IP training; reasons were older students, more experience in health care; less attitudes toward accepting change, and hidden culture that contradicts interprofessional message Grading verses not grading of assignment created tension in level of commitment to assignments and productivity Staff resistance increases if there is lack of clarity regarding work, motivation of staff and misalignment of structures, workload and loss of identity Patients may prefer primary care provider rather than IP team approach Sustaining long-term effects from educational venues into clinical practice

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
