# Peer review of "An Integrative Review of Interprofessional Collaboration in Health Care: Building the Case for University Support and Resources and Faculty Engagement"

_healthcare, 2020, doi:10.3390/healthcare8040418_

Round 1

Reviewer 1 Report

I enjoyed reading this work about an important issue in higher education. I have the following thoughts and suggestions for the authors:

  1. Lines 89-91. To me this suggests that you were looking to prove the value - was this the reality - were you seeking this as a way of proving the value so that you are potentially accepting and putting forward that you were positive towards the value of these approaches? Or were you just investigating the phenomenon? This is not quite consistent with the suggestion in the Problem identification below. As a reader I need to know whether you are seeking to prove and therefore accepting a 'bias' or influence in this or merely seeking to explore.
  2. Line 132 - Why 20 points as a cut off out of 42? Explain for the reader why you took that as a cut off?
  3. Line 153-155 I would re-write this to say ten were in the United States, three in the UK, two in Canada and one each in Germany, Sweden and one across Europe.
  4. Line 228 - remove 'the' before 'exploring'.
  5. Your review provides a significant amount of data which you have highlighted in different sections in the Tables, for example Changes Related to Knowledge. I am not convinced that the Discussion section does this justice. I would encourage this section to be revisited and expanded because at present the reader has to extrapolate from your tables the links and where this narrative has been formed from. I think this would then help to facilitate the section implications section more. I also feel that the section headed implications is potentially more recommendations rather than the implications of what you have found, which goes back to my original comment about whether you were looking to find support for this so you could make these implications of the research become recommendations. 
  6. I would think some limitations to the approach could be added for the reader?

Author Response

Author responses to reviewers are attached.

Reviewer 2 Report

This is an interesting, well-written paper on an important topic, which I enjoyed. I do have some methodological queries however, as below.

Page 2, line 46: You state “essential behavioral combination of knowledge, attitudes, skills and values make up collaborative practice” – but “knowledge, attitudes, skills and values” are not behavioral, they are cognitive, hence why do these make up an essential _behavioral_ combination?

Page 3, line 110: your search keys and engines seem fine but given such a systematic search I would have expected to see a PRISMA flow chart describing the inclusion/exclusion process. Also what were your inclusion and exclusion criteria – this seem rather vague since you went from 216,885 papers to 18, so exclusion must have been strict? You mention that fewer papers included undergraduate IPE, and I notice few of your included papers involve simulation – but two papers I am familiar with which do include undergraduate IPE participants and simulation which have not been included in your review (but which appear to fit your criteria and time period) are:

Martini, N et al. Designing and Evaluating a Virtual Patient Simulation—The Journey from Uniprofessional to Interprofessional Learning. Information 2019, 10, 28.

Webster, C.S. et al. Advanced Cardiac Life Support Training in Interprofessional Teams of Undergraduate Nursing and Medical Students Using Mannequin-Based Simulation. Med.Sci.Educ. 28, 155–163 (2018). https://doi.org/10.1007/s40670-017-0523-0

A better description of your inclusion/exclusion would help me understand why. Also why did you choose 1995 as your start point? Any particular reason?

Page 4, line 132: Is there a particular reason for the evidence threshold of 20 points for inclusion in your study? Is this a significant level?

Page 4, line 143: “structure speaks to the setting” – this personifies structure, perhaps rephrase.

Page 15, Implications: I like your four implications. But its less clear to me how these fit with the tables of data presented from your review. I presume these implications or recommendations are evidence-based on the results of your review?

Page 17, Conclusion: Your conclusion seems very short and abstract. It would be better to get some of the specific recommendations from your Implications section into the conclusion to emphasize the take home messages. It would also be good to get these take home messages into the abstract.

END

Author Response

Responses are attached
